# The Preparation and Structure Analysis Methods of Natural Polysaccharides of Plants and Fungi: A Review of Recent Development

**DOI:** 10.3390/molecules24173122

**Published:** 2019-08-28

**Authors:** Yan Ren, Yueping Bai, Zhidan Zhang, Wenlong Cai, Antonio Del Rio Flores

**Affiliations:** 1College of Pharmacy, Southwest Minzu University, Chengdu 610225, China; 2Tianjin Institute of Industrial Biotechnology, Chinese Academy of Sciences, Tianjin 300308, China; 3Department of Chemical and Biomolecular Engineering, University of California Berkeley, Berkeley, CA 94720, USA

**Keywords:** natural polysaccharides, plants and fungi, extraction, isolation, purification, structure analysis

## Abstract

Polysaccharides are ubiquitous biomolecules found in nature that contain various biological and pharmacological activities that are employed in functional foods and therapeutic agents. Natural polysaccharides are obtained mainly by extraction and purification, which may serve as reliable procedures to enhance the quality and the yield of polysaccharide products. Moreover, structural analysis of polysaccharides proves to be promising and crucial for elucidating structure–activity relationships. Therefore, this report summarizes the recent developments and applications in extraction, separation, purification, and structural analysis of polysaccharides of plants and fungi.

## 1. Introduction

Polysaccharides are polymers comprising monosaccharide residues connected by glycosidic bonds. The structures of polysaccharides generally contain linear or branched side chains, and the molecular weight is distributed over the range of tens of thousands to millions [1]. Polysaccharides, derived from multiple natural resources such as plants, animals, bacteria, fungi, algae, arthropods, etc. (Figure 1), are not only an important component of energy and structural components but also serve a variety of biological functions. Many of these functions include signal recognition, intercellular connection, immune system regulation, blood coagulation, and pathological inhibition [2]. Recently, studies have increasingly shown that polysaccharides have extensive pharmacological activity such as anti-tumor, anti-virus, anti-inflammatory, cardiovascular protection, and anti-mutation effects [3,4].

Polysaccharides with multiple functional groups (hydroxyl, amino, and carboxylic acid groups) could be further modified for bio-medicinal applications, such as for vaccine adjuvants, drug carriers, tissue engineering scaffolds, et cetera [5]. Recently, prominent research has contributed to progress in exploiting the pharmacological, activity of natural polysaccharides, as well as polysaccharides-based biomaterials for various applications such as tissue engineering and regenerative medicine [6,7,8]. Nevertheless, polysaccharides are still a mystery to us. In the first place, the structure–activity relationship is unclear. Secondly, polysaccharide detection and quantification has not been standardized. Although the synthesis of polysaccharides has undergone significant developments, limited by their structure complexity, it is still challenging and time consuming to pursue the synthetic route due to the lack of commercial automated synthesizers and difficulties in controlling the stereochemistry of the glycosidic linkages [9]. Figure 2 shows the number of studies on the synthesis of polysaccharides and the extraction of natural polysaccharides reported in the last five years. Compared with synthetic polysaccharides, the data demonstrate that the extraction of natural polysaccharides is still the mainstream approach to obtaining polysaccharides. Therefore, effective separation and purification of polysaccharides are the key steps outlining the consensus. Moreover, they are the preconditions for structural identification and study of the structure–activity relationship as well. Although earlier overviews on the bioactivity and the general methods of isolation of polysaccharides have been reported by Lei [10], herein, we review the more recent progress in the development of holistic processes for polysaccharide acquisition, including extraction, separation, purification, and structure analysis. This review focuses specifically on polysaccharides derived from plant and fungus resources in the hope of promoting further studies on natural polysaccharides.

## 2. Extraction of Polysaccharides

Polysaccharides are polar molecules that are often soluble in water but insoluble in organic solvents. Therefore, water is currently the most common extraction solvent and the base solvent of other extraction methods, including acid and alkali extraction [11] and the enzymolysis method [12]. Excluding polysaccharides from exudates, the main principle of polysaccharide extraction is to break the cell wall from the outside to the inside under mild conditions, thus avoiding polysaccharide denaturation [13]. Based on this approach, below, we summarize several common methods for polysaccharide extraction developed in recent years.

### 2.1. Ultrasonic Extraction

The ultrasonic extraction method based on the ultrasonic wave cavitation is used to break cell walls and accelerate the dissolution of organics in cells, thus improving the yield of polysaccharides [14]. Zhu et al. used ultrasound waves to extract polysaccharides from *Polygonum multiflorum*, achieving a maximum extraction rate of 5.49%. After separation and purification, both neutral and acidic components showed high anti-tumor activity [15]. Recent research suggests that the ultrasonic extraction method could significantly improve the rate of polysaccharide dissolution, whereas prolonged exposure to ultrasound may change the advanced structure of polysaccharides and affect the biological activity.

### 2.2. Microwave Extraction

The principle of the microwave extraction method is that, when the cell absorbs microwave energy, the intracellular pressure will increase, leading to cell rupture and causing the active components to flow into the solvent [16]. The microwave-assisted extraction method can improve the yield of polysaccharides significantly but save energy and time. For instance, the yield of polysaccharides from *Cyphomandra betacea* was determined to be 36.52% under the optimal conditions [17]. However, a rapid temperature spike may change the molecular mass distribution and the structures of the thermally unstable polysaccharides.

### 2.3. Supercritical CO_2_ Extraction

CO_2_ in the supercritical state can be used to selectively extract components of a mixture due to differences in their polarity, boiling point, and molecular weight based on the relationship between the solubility in the supercritical fluid and the density. In this system, the use of appropriate entrainers and modulators (e.g., methanol and ethanol) can also improve the rate of extraction of polysaccharides [18,19]. Supercritical CO_2_ fluid has good solvency and high efficiency in analytical procedures [18].

### 2.4. Subcritical Water Extraction

Subcritical water, also known as pressurized hot water, refers to water heated above the normal boiling point of 100 °C but below the high critical temperature of 374 °C at a given pressure while remaining in its liquid state. Subcritical water extraction has been successfully applied to extraction of the active components of natural products [20]. Yang et al. compared the extraction rate of *Lycium barbarum* fruit polysaccharides by using hot water extraction, ultrasonic extraction, subcritical water extraction, and ultrasound combined with subcritical water extraction, respectively. The results showed that the extraction rate achieved with ultrasound combined with subcritical water extraction was 6.4% [21]. Although the application of subcritical water extraction to natural polysaccharides is still in the stage of basic research and has not reached the level of industrial production, this method can be used to extract polysaccharides with better quality and maintain higher activity.

### 2.5. Other Extraction Methods

Dynamic high-pressure micro-jet technology is an emerging technology combining shear, high-frequency vibration, cavitation, and instantaneous pressure drop, with a maximum pressure of 200 MPa. Although this method has the obvious advantages of a short extraction time and high extraction rate [22], it is possible to break the polysaccharide chain and change its original characteristics.

Due to the opaque heating from infrared radiation, this method offers high permeability, which is conducive for effective dissolution of the cell components. Infrared radiation is increasingly applied to the auxiliary extraction of active components from natural products [23]. This method has the advantages of a short extraction time, low temperature operation, free irradiation, and low cost. Considering the diversity and the complexity of the extracted polysaccharides, such as snow chrysanthemum polysaccharides, increasing attention has been focused on the cooperative utilization of multiple technologies, where each method is exploited to maximize the effectiveness [24]. Appropriate extraction methods are helpful for retaining the inherent properties of polysaccharide molecules and preventing chain fracture and changes in the spatial conformation. The various methods for extraction of natural polysaccharides are presented in Figure 3 based on intuitive organization of the basic strategies of the extraction methods.

## 3. Separation and Purification of Polysaccharides

The polysaccharides extracted from natural products contain many impurities, such as inorganic salts, protein, lignin, etc. It is difficult to evaluate the structure–activity relationship of crude polysaccharides, thus certain measures are required to separate the crude polysaccharides. The mixture of different polysaccharides must be purified into a single polysaccharide component with the same degree of polymerization and spatial conformation.

### 3.1. Separating Impurities from Polysaccharides

#### 3.1.1. Protein Removal

Proteins and polysaccharides are complex hydrophilic biopolymers with diverse structures. The separation of proteins from crude polysaccharides is a key step in the separation and the purification of polysaccharides. The conventional methods for this process are the Sevag method and the trichloroacetic acid method based on the principle that the reagent denatures and precipitates proteins instead of polysaccharides. However, the Sevag method is complicated and time-consuming [25]. Apart from the direct addition of a common protease for protein removal, protease secretion by microorganisms is also used; for example, *Saccharomyces cerevisiae* can remove proteins [26]. In order to compensate the deficiency of a single method, the Sevag-enzyme combination could effectively reduce the loss of polysaccharides.

#### 3.1.2. Pigment Removal

The phenolic compounds present during the extraction of natural polysaccharides produce pigments, as is known, where the pigments affect the chromatographic analysis and impede accurate identification of the polysaccharides due to oxidation. In particular, the crude polysaccharides from animals are darker than those from plants. Decolorization methods usually include the resin method, the activated carbon method, and the hydrogen peroxide oxidation method. Specifically, the ion exchange resin or the adsorption resin with the advantages of a high decolorization rate and stable characteristic group structures after decolorization have commonly been used in recent years [27,28].

### 3.2. Purification of Polysaccharides

After removal from the cell, a polysaccharide is not a single molecule but a mixture with different degrees of polymerization. Therefore, deep purification is the basis for studying the relationship between the structure and the biological activity. Based on the separation mechanism, purification techniques can be divided into three categories: physical purification, chromatographic purification, and chemical precipitation. In recent years, a single purification method has been seldom used, and the combination of several separation methods and several devices has been employed to improve the purification results.

#### 3.2.1. Precipitation

Fractional precipitation is suitable for polysaccharides with large differences in solubility and molecular weight [29,30,31]. In addition, the long chain quaternary ammonium salt chemical precipitation method and the metal complex method are also applied. The principles of precipitation are shown in Table 1.

#### 3.2.2. Chromatographic Separation

Column chromatography is an efficient purification method for the separation and the purification of natural components. Based on the physicochemical properties of the target substance, the most suitable stationary phase and mobile phase are selected for achieving high yield of the target substance. According to the working principle of the stationary phase filler, column chromatography can be divided into cellulose column chromatography, ion exchange column chromatography, gel column chromatography, and affinity column chromatography. In recent years, dicthylaminoethyl (DEAE)-cellulose anion exchange column chromatography and gel column chromatography have been used in tandem to purify polysaccharides [32,33,34,35].

##### 3.2.2.1. Anion-Exchange Column Chromatography

The separation of polysaccharides by anion-exchange column chromatography is generally used as the primary stage for the purification of crude polysaccharides [36] and is based on the principles of adsorption and partition chromatography. For the ion exchange resin, chromatographic separation is achieved by reversible exchange, electron–dipole interaction, or adsorption among surface charged groups of the stationary phase, ions of the sample, and ions of the mobile phase. Methods employing exchange media containing DEAE-cellulose are widely used, where DEAE-Sepharose and DEAE-dextran gel [37,38] are suitable for separating various acidic, neutral, and mucopolysaccharides. Acidic polysaccharides can be adsorbed on the exchanger at pH 6, whereas neutral polysaccharides cannot. Moreover, the different acidic polysaccharides can be eluted by using a buffer with the same pH but different ionic strength. In addition, if the column used is alkaline, the neutral polysaccharides can also be adsorbed, where the adsorption capacity depends on the number of acidic groups in the molecule.

##### 3.2.2.2. Gel Column Chromatography

Different polysaccharides are separated by gel column chromatography (GPC) based on the molecular sieve action of the gel (porous network structure in three dimensions), which depends on the speed of motion of the polysaccharides with different molecular sizes and shapes in the chromatography column. Before purification, a gel with small voids can be used to remove impurities such as small molecules and inorganic salts. Gels (for example, dextran gel, polyacrylamide gel, and agarose gel) are commonly used as the stationary phase, and deionized water or dilute salt solution are used as the eluent. To avoid tailing, the ionic strength of the eluent should be greater than 20 μM. Different gels are appropriate for polysaccharides of different molecular masses. Therefore, the specific gel column should be selected according to the relative molecular mass of the target polysaccharide [39].

In most cases, anion-exchange chromatography is used in the first step, followed by gel column chromatography, as shown in Figure 4 [40,41,42]. This combined method is simple and mild but may be potentially effective for the separation of viscous polysaccharides that tend to undergo adhesion.

#### 3.2.3. Other Polysaccharide Purification Methods

Depending on the size, the shape, and the charge characteristics of the polysaccharides, the migration speed of the polysaccharide under the action of an electric field varies, and preparation zone electrophoresis can be used to separate various polysaccharides. In addition, homogenous polysaccharides can be separated by ultracentrifugation. A comparison of the main separation and purification methods mentioned in this section is presented in Table 1.

Generally, it is difficult to obtain pure polysaccharides by one method; thus, the combination of multiple methods is needed to achieve efficient separation of polysaccharides. The scope and the order of application of each method should also be considered in this process.

Unfortunately, there has been no major breakthrough in the purification methods and materials in recent years. The discovery of new methods and materials is required for elucidating the structural characteristics of polysaccharide molecules. Therefore, an in-depth study on the structure of existing polysaccharides can promote the innovation of polysaccharide purification.

## 4. Structural Analysis of Polysaccharides

At present, most researchers pay more attention to the primary structures rather than the advanced structures of polysaccharides. Therefore, a feasible technology for analyzing the structure of polysaccharides is urgently needed.

### 4.1. Determination of Primary Structure

#### 4.1.1. Detection of Polysaccharide Homogeneity

Homogeneity of a polysaccharide suggests that the components of the purified polysaccharide are homogeneous or pure. In order to lay a good foundation for subsequent structural identification, determination of the purity of polysaccharides is a necessary process to ensure the uniqueness of the polysaccharides.

##### 4.1.1.1. Chromatography

Chromatography is currently the most common method for identifying the homogeneity of polysaccharides, for example, gel permeation chromatography and HPLC combined with differential refractive index detector (RID). The method can be used to detect the homogeneity of the polysaccharide as well as calculate the molecular weight of the polysaccharide [43].

Polysaccharides of different masses can move at different speeds in the gel column; thus, we can use an appropriate flow rate to elute different components of the polysaccharide sample. Specifically, the absorbance curve is plotted by using the tube number and the absorbance measurement as the ordinates. In this method, if a single and symmetrical peak appears, the component is usually considered to be a homogeneous polysaccharide [44,45]. The polysaccharide content can also be measured by the phenol-sulfuric acid method [46].

##### 4.1.1.2. Other Methods

Polyacrylamide gel electrophoresis and cellulose acetate membrane electrophoresis in conjunction with GPC are also commonly used for determining the molecular weight of polysaccharides. The two methods are carried out simultaneously to further confirm the purity of polysaccharides [47,48,49].

#### 4.1.2. Determination of Molecular Weight of Polysaccharides

The physicochemical properties and the pharmacological activity of polysaccharides are closely related to the molecular weight [50,51,52,53].

Polysaccharides with similar structures generally have different molecular weights. The average molecular weight is obtained by dividing the molecular mass of *n* polymers that make up the polysaccharides into *n*, and the average molecular weight is calculated based on the relative molecular weight [54]. A degree of dispersion (Mw/Mn) of unity is indicative of uniformity, and a degree of dispersion greater than unity reflects a wide molecular weight distribution [55].
(1)$$M_{n}=\frac{\sum_{i}M_{i}N_{i}}{\sum_{i}N_{i}}$$
(2)$$M_{w}=\frac{\sum_{i}M_{i}^{2}N_{i}}{\sum_{i}N_{i}}$$
where *N_i_* is the number of molecules of molecular weight *M_i_.*

There are many methods for determining the molecular weight of polysaccharides. Among them, the most commonly used method is HPLC.

##### 4.1.2.1. High Performance Liquid Chromatography

Most of the current studies employ high-performance gel permeation chromatography (HPGPC) or high-performance exclusion chromatography (HPSEC). These techniques offer the advantages of speed, high resolution, and reproducibility and can simultaneously detect the homogeneity of polysaccharides [56,57]. The most commonly used columns are μ-Bondagel, TSK, Sephadex, Superose, etc. The mobile phases include water, buffer, or aqueous organic solvent. The detectors include refractive index refractometers, evaporative light scattering, multi-angle excitation diffuser, etc. [36,58].

##### 4.1.2.2. Gel Permeation Chromatography

In a gel column of a certain length, the polysaccharide molecules are separated according to the relative molecular weight [59]. In some practical applications, GPC is combined with a multiangle laser scattering detector (MALLS) [60]. This method does not require calibration with a reference material and has high accuracy and precision.

##### 4.1.2.3. Mass Spectrometry

MALDI-TOF-MS is often used to analyze biological macromolecules. The collision-induced cleavage, electron transfer cleavage, electron capture cleavage, post-source decay, and other post-source cleavage techniques are not only used for determination of the molecular weight of polysaccharides but also for identification of the structural fragments [61]. In the measurement process, the matrix and the sample concentration are selected according to the structure of the polysaccharide to achieve the desired result.

##### 4.1.2.4. Other Methods

Solutions of high molecular weight polysaccharides generally have higher viscosity. Therefore, the viscosity method can be used to determine the molecular weight of polysaccharides. In practice, there are some uncertainties in the method, because determination of the viscosity is generally influenced by many factors such as molecular weight and molecule shape.

#### 4.1.3. Monosaccharide Composition

In general, natural polysaccharides are composed of different monosaccharides. Hydrolysis is commonly used to analyze the monosaccharide components of polysaccharides. The determination of the monosaccharide composition is helpful for predicting the core structure of the polysaccharide main chain and studying the physicochemical properties of polysaccharides. In recent research, polysaccharides were hydrolyzed to monosaccharides or subjected to complete hydrolysis for further analysis of the monosaccharide composition using various chromatographic methods.

Acid hydrolysis is the first step in the analytical process, where the process differs based on the type of polysaccharide. Trifluoroacetic acid is commonly used for hydrolyzing neutral hexose, pentose, deoxyhexose, etc. The excess acid is removed by total water distillation, and the hydrolytic product is reduced with NaBH_4_. The reduced polysaccharide is acetylated in a boiling water bath using 1:1 pyridine-acetic anhydride and is then analyzed by GLC, which can be used to determine the types of monosaccharides as well as to quantitatively assess the proportion of monosaccharides [62].

##### 4.1.3.1. High-Performance Liquid Phase Chromatography

HPLC is a high-frequency method for detecting the composition of monosaccharides in polysaccharides [63]. First, the polysaccharide is hydrolyzed to monosaccharides, and the monosaccharides are then chemically derivatized; for example, a fluorescent group is introduced to increase the sensitivity of the detection. The commonly used derivative reagent is 1-phenyl-3-methyl-5-pyrazolone (PMP) [36].

##### 4.1.3.2. High-Performance Capillary Electrophoresis

In high-performance capillary electrophoresis (HPCE), the polysaccharide is labeled with a reagent with an acidic group and then detected by a laser induced fluorescence detector (LIFD) and analyzed by HPCE. It has been reported that the polysaccharide is hydrolyzed to monosaccharides by an acid and is derivatized by PMP [64]. The sample is then assayed by the apparatus. The mole percentage of monosaccharides can be calculated using the peak area [65,66].

##### 4.1.3.3. Ion Chromatography

High-performance anion exchange chromatography combined with pulsed amperometric detection (HPAEC-PAD) is a new technology that has been used for analyzing polysaccharide samples in recent years [67,68,69]. The monosaccharides and the oligosaccharides from hydrolyzed polysaccharides can be dissociated into anions by elution at pH > 12, where the polysaccharide is exchanged and distributed on the high-efficiency anion exchange resin and detected by a pulsed amperometric detector. The device is equipped with an electrochemical detector consisting of an Au and an Ag working electrode reference electrode, respectively. Elution is performed with a solution of pure water with sodium hydroxide and sodium acetate for step-gradient elution. By comparison with standard samples, for example, glucose, fructose, galactose, mannose, and other monosaccharides, the monosaccharide composition can be determined [70,71]. This method is being increasingly applied, as it allows direct analysis of the hydrolytic products without sample derivation.

##### 4.1.3.4. Formation of Methyl Glycosides

In analysis of the monosaccharide composition using GC-MS, the sample was dissolved in a 10 mg/mL of sodium borohydride (NaBH_4_) solution. After that, glacial acetic acid was added to neutralize the excess NaBH_4_ until no bubbles appeared. Then, methanol was added, the solution was evaporated to dryness, and the sample was put into a vacuum freeze dryer overnight. The sample was heated in the oven at 100 °C for 15 min to remove the residual water fully and then converted to acetylated products. The acetylated product was dissolved into chloroform [72]. When lipopolysaccharide sulfates are analyzed with methanol, the sulfuric acid produced may destroy the galactose, the glucuronic acid, and other reactants. Therefore, an appropriate amount of barium sulfates should be added to remove the sulfuric acid during the process.

Notably, the polysaccharide structures may contain uronic acid residues; thus, the content of uronic acid must be measured separately in the analysis of the monosaccharide composition.

As described above, the chromatographic method has a significant impact on the separation and the purification of natural polysaccharides, homogeneity detection, molecular weight determination, and monosaccharide composition analysis. Table 2 presents a summary of select studies using chromatography for the analysis of natural polysaccharides conducted in the last five years. Furthermore, we also summarize the studies on monosaccharide composition, molecular weight, main structure, and biological activity of part of natural polysaccharide work performed in the last five years (Table 3).

#### 4.1.4. Analysis of Chain Structure of Polysaccharides

##### 4.1.4.1. Methylation Analysis

The methylation analysis methods include the dimethyl sulfate method, the Hakomori method, and the Ciucanu method (1984), among which the Ciucan method (1984) is most frequently utilized [109,110]. The definite mechanism of the oxidation process, including ways to avoid the oxidation of the carbohydrate, was investigated by Ciucanu in 2003 [111]. Furthermore, Ciucanu introduced a novel method in which per-*O*-methylation of neutral carbohydrates was carried out in one step by adding dimethyl sulfoxide, powdered sodium hydroxide, and methyl iodide directly to an aqueous sample [112]. After hydrolysis, the hydroxyl group of the methylated monosaccharide was released and acetylated, and the resulting methyl derivative of each monosaccharide was analyzed by GC-MS. The most important thing in this process is to ensure the completeness of methylation, which is detectable by infrared spectroscopy [113]. If the polysaccharide contains uronic acid, reduction is required before methylation.

Currently, a relatively simple process for assessing the monosaccharide linkage by methylation is to first methylate the polysaccharides, then recover the product by dialysis and freeze-drying, hydrolyze with trifluoroacetic acid, and acetylate after borohydride reduction to produce partially methylated monosaccharide derivatives such as methylated aldonic acetate (PMAA). The derivatives are then analyzed by GC-MS, and the spectra are compared with those in the Carbohydrate Research Centre Spectral Database-PMAA to determine the composition and the connectivity [66].

##### 4.1.4.2. Periodate Oxidation

Because the *o*-diol and the *o*-triol structures in polysaccharides are easily oxidized by hyperiodic acids (or the salts thereof) to generate the corresponding polysaccharide aldehydes, formaldehyde, or formic acid, the position of connection of various monosaccharides in polysaccharides can be determined by measuring the consumption of periodate and the production of monosaccharides. As one example, the structure of a polysaccharide is analyzed under controlled conditions, while glucose is analyzed by periodic acid oxidation. Generally, when the pH is controlled between 3 and 5, polysaccharides react with the smallest amount of sodium periodate [114]. Therefore, the periodic acid (or salt) oxidation method for determining the structures of polysaccharides must be combined with methylation analysis and Smith degradation to achieve accurate and reliable results [115].

##### 4.1.4.3. Smith Degradation Reaction

Smith degradation involves acid hydrolysis or partial acid hydrolysis after reduction of the product of the oxidation of hyperiodic acid. Various erythritol glycosides and glycerol glycosides are obtained. Studying the structure of these monoglycosides or oligoglycosides is useful for elucidating the partial joining sequence and the position of attachment of the monosaccharides in the polysaccharide. The oxidation product is reduced to a stable polyhydroxy compound by using a boron hydride compound, and the hydrolysis product is identified by gas chromatography after acid hydrolysis to infer the position of the glycosidic bond [116].

##### 4.1.4.4. Enzymatic Hydrolysis

Specific enzymes can hydrolyze specific glycosidic bonds, hydrolyze structurally complex polysaccharides into relatively simple structural fragments, and be employed in structural analysis. Polysaccharide hydrolases include α-amylase, β-1,4-xylanase, and β-1,4-mannanase. In the structural study of polysaccharides, after the monosaccharide composition and the attachment positions of a polysaccharide are determined, enzymatic hydrolysis is carried out using a suitable enzyme, and specific enzymatic hydrolysis is carried out to generate small fragments. Various chromatographic techniques are used to analyze each fragment [117]. The application of carbohydrase makes structural evaluation of polysaccharides easier and can greatly reduce the difficulty in elucidating the polysaccharide structure.

##### 4.1.4.5. Infrared Spectroscopy

Infrared spectroscopy is mainly used to determine the configuration of glycosidic bonds. For example, a distinct broad peak at 3500−3000 cm^−1^ is assigned as the O-H bond stretching vibration peak. A peak in this region can be used to judge whether the polysaccharide is completely methylated. Peaks at 3600−3200 cm^−1^, 3000−2800 cm^−1^, and 1200−1000 cm^−1^ are characteristic absorption peaks of polysaccharides [118].

##### 4.1.4.6. Raman Spectroscopy

Raman spectroscopy is mostly used for detecting the vibrations of polysaccharide molecules, atomic non-polar bonds, and isomers. It is reported that the C-O-C vibration involving α-d-(1→4) linkages mainly contributes to bands in the regions of 960 cm^−1^ and 920 cm^−1^ [119]. Four absorption bands of the glycosidic bonds are present in the Raman spectra (Table 4).

##### 4.1.4.7. Nuclear Magnetic Resonance Spectroscopy (NMR)

NMR mainly provides confirmation of the glycosidic bonds in the polysaccharide structure and the number of monosaccharides in the repeating structure. NMR simplifies the polysaccharide analysis steps and provides information on the polysaccharide structure, including monosaccharide identification, alpha or beta anomeric confirmation, glycosidic linkage type, and the repetitive unit sequences of the polysaccharide chain. For example, the end-substrate signal of the ^1^H-NMR spectrum of polysaccharides appears between 4.5 and 5.5 ppm. Compared with protons at other positions, the end-matrix is located relatively downfield, and the corresponding signal peaks (two coupling) are well separated, corresponding to a single hydrogen doublet. Except for the anomeric signals, all the peaks of the other protons in the polysaccharide appear at 3.0−4.0 ppm, though extensive overlapping of the peaks makes analysis very difficult, especially for hetero-polysaccharides. Most of the peaks in the ^13^C-NMR spectrum of polysaccharides appear between 60 and 110 ppm. Compared with ^1^H-NMR spectrum, ^13^C-NMR has a wide range of displacement values and only a few of the signals overlap.

The end-group carbon signal is very important for inferring the type of polysaccharide residue in the polysaccharide molecules. In addition, integration of the terminal signal in the fully decoupled ^13^C-NMR spectrum can be used to examine the type of polysaccharide residue. At present, 2D NMR (for example, HSQC and HMBC) is the main method for analyzing monosaccharide structures, glycosidic bond types, and so on [120]. HMBC can correlate hydrogen protons with remote carbons for hydrocarbon assignment and analysis of the structural skeleton.

Additionally, some biotechnologies such as lectin recognition and fluorescent labeling have also been applied to the structural analysis of polysaccharides. However, due to the complicated structures of polysaccharides, an appropriate experimental method is selected according to a particular structure. A summary of the methods commonly used for determination of the primary structure of polysaccharides is presented in Figure 5 and Table 5.

### 4.2. Conformational Analysis of Polysaccharides

Polysaccharides are highly hydrophilic because their structures contain a large number of hydroxyl groups. Polysaccharide molecules can form three-dimensional network structures via hydrogen bonds, van der Waals forces, covalent bonds, etc. Moreover, the polysaccharide chain has a large degree of freedom and flexibility, and inter-chain association is facile. These features make the spatial conformation of polysaccharides very complex. Compared with primary structures, there is little research on the advanced structures due to the more challenging nature of the latter. Studies have increasingly demonstrated that the bioactivity of polysaccharides is not only related to the primary structures but is also more strongly influenced by the advanced structures [121,122]. Therefore, analyses of the advanced structures of polysaccharides are of great significance. The main methods of studying the advanced structure of polysaccharides are introduced below.

#### 4.2.1. Scanning Electron Microscopy (SEM)

SEM is widely used to observe the microscopic structure of the surfaces of natural polysaccharides. SEM can be used to acquire high-resolution images, which provide a sense of three-dimension and reality. Due to the high resolution and the large depth of field of SEM, the rough surface structures of polysaccharides with large fluctuations can be directly observed [123]. Furthermore, SEM can be used for structural characterization of purified single polysaccharides as well as for intuitive tracking for the separation and the purification of structurally stable polysaccharides. In addition, the diversity and the specificity of the three-dimensional shapes of different polysaccharides can provide enlightenment for the identification of species or the search for markers. Using this method, our laboratory observed the polysaccharides structure from the bamboo shoot shell of *Pleioblastus amarus* as shown in Figure 6.

#### 4.2.2. Atomic Force Microscopy (AFM)

AFM is a new surface analysis device that can be used to analyze the three-dimensional structure of biological macromolecules on the nanoscale. It is also used to study the surface morphology of high molecular weight polymers [124]. In AFM, a constant force is maintained between the control sample and the probe. When the probe scans the surface of the sample, its motion trajectory can be recorded and converted to a complete three-dimensional image. AFM has extremely high resolution, thus accurate microscopic images of the sample surface can be obtained. Wang et al. analyzed the structure of polysaccharides from *Ophiopogon japonicas* by AFM. The results are consistent with those of NMR analysis, indicating that ultrasonic extracted polysaccharides purified from *Ophiopogon japonicas* (POJ-U1a) contain branches. The width of the branches was greater than that of single polysaccharides, indicative of molecular aggregation [125].

#### 4.2.3. X-ray Diffraction

X-ray diffraction is a usual method for studying crystal structures. The orientation and the intensity of X-ray diffraction are closely related to the crystal structure. In the application to polysaccharides, X-ray diffraction can be used to estimate the symmetry and the spiral parameters. However, most polysaccharides cannot form single crystals, thus sample processing is required. In addition, X-ray fiber diffraction can be combined with stereochemistry determination and computer simulation techniques for adequate determination of the conformation of polysaccharides.

#### 4.2.4. Circular Dichroism Chromatography

Circular dichroism chromatography, a spectroscopic method for determining molecular asymmetric structures, has mature applications in the advanced structural analysis of proteins [126]. Transformation of the advanced structure of the polysaccharide changes the polarization, the static force, and the orientation of the chromophores such as amino, acetyl, and carboxyl groups. The circular dichroism changes significantly in the range of 200−400 nm [127]. Chain conformation analysis and circular dichroism strongly demonstrated that shielding by negatively charged components promoted transition of the polysaccharide chain towards more rigid conformation after sulfonation of *Artemisia sphaeroides* polysaccharides [128].

## 5. Conclusions

To exploit the prominent application prospects of natural polysaccharides in the development of pharmaceutical materials and functional foods, growing efforts have been devoted to improving separation and purification technology for obtaining natural polysaccharides. Although the methods for analyzing the advanced structures of polysaccharides are not accurate and comprehensive, and the structure–activity relationship of polysaccharides cannot be confirmed at the molecular level, presently, we believe that, with the development of technology, interdisciplinary integration will provide more new ideas and methods for polysaccharide identification. Natural polysaccharides are prospectively the most precious wealth that nature has given us.

## Figures and Tables

**Figure 1 molecules-24-03122-f001:**
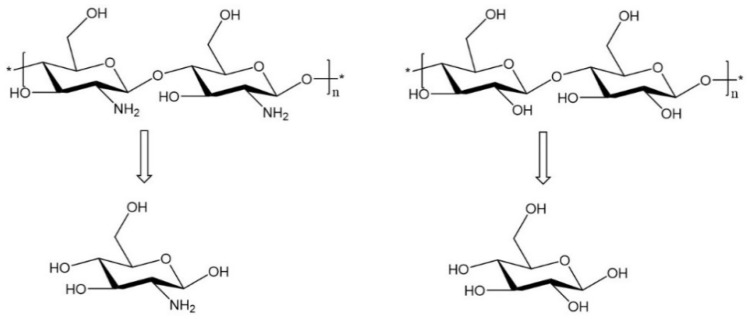
Structures of animal and plant polysaccharides (Left: chitosan; right: cellulose).

**Figure 2 molecules-24-03122-f002:**
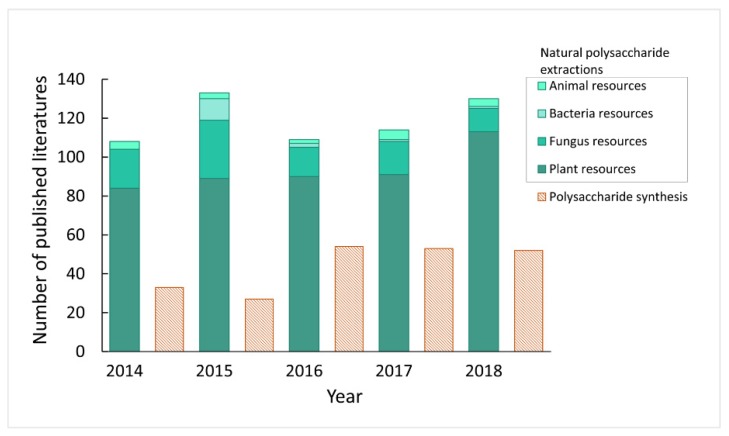
Literature analysis of natural polysaccharide extractions and polysaccharide synthesis with the statistical data coming from the web of science.

**Figure 3 molecules-24-03122-f003:**
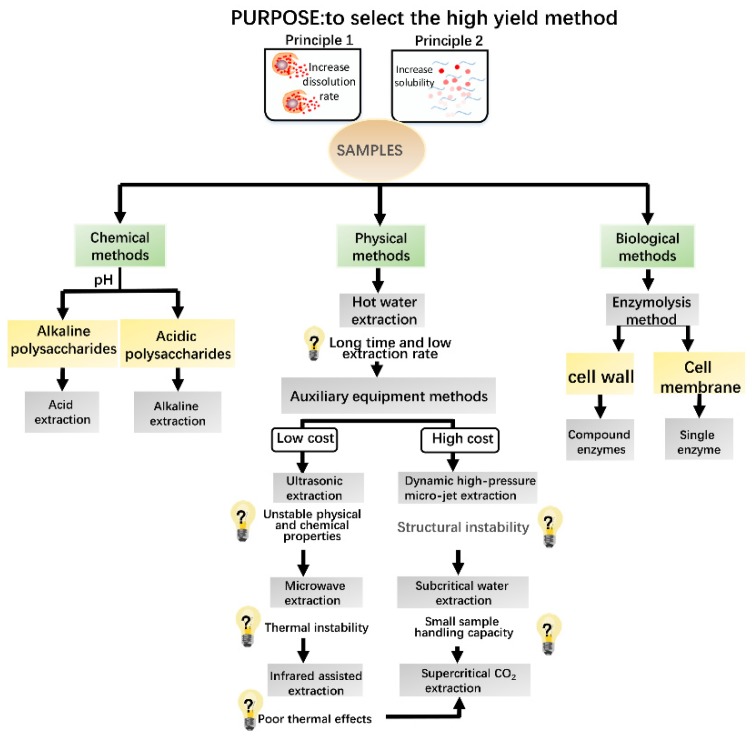
Strategic map for selection of natural polysaccharide extraction methods.

**Figure 4 molecules-24-03122-f004:**
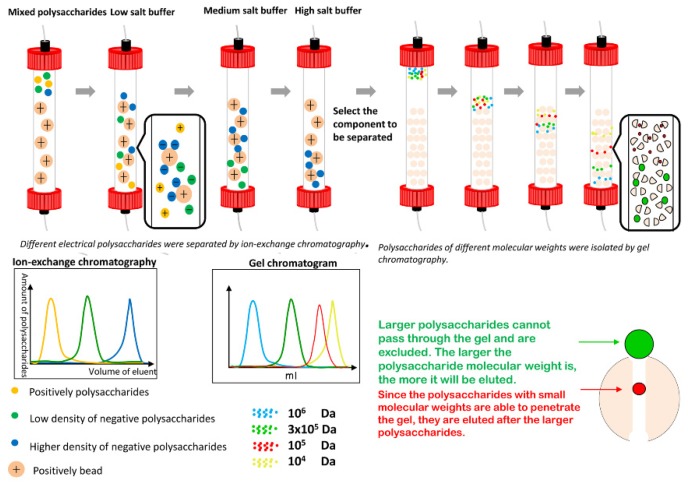
Schematic diagram of ion exchange and gel column chromatography.

**Figure 5 molecules-24-03122-f005:**
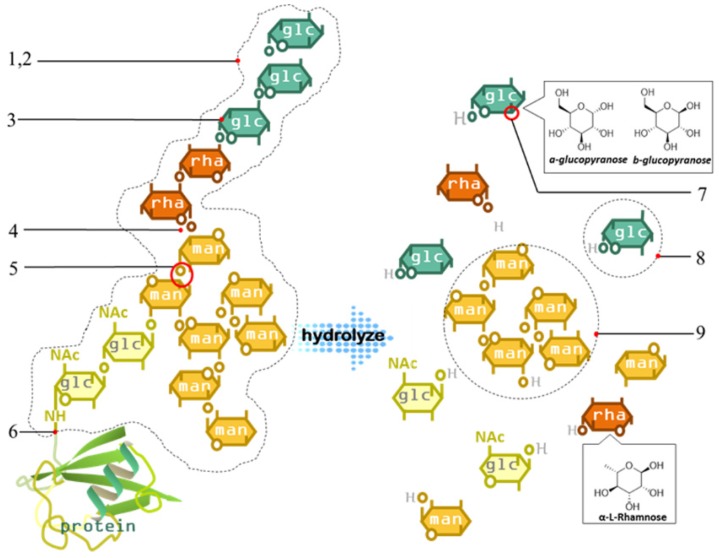
Analysis of the primary structure of polysaccharides (icon Table 5).

**Figure 6 molecules-24-03122-f006:**
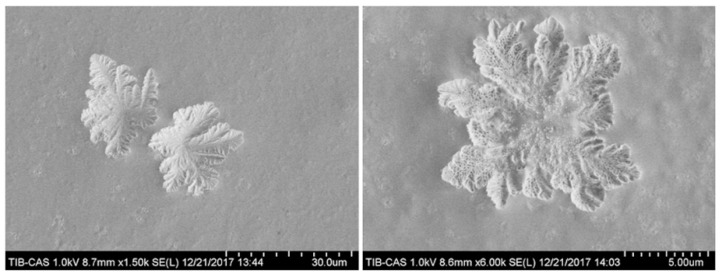
Photomicrographs of polysaccharides as recorded by SEM (these three polysaccharides came from the bamboo shoot shell of *Pleioblastus amarus* extraction in our laboratory).

**Table 1 molecules-24-03122-t001:** Summary of methods of purification of polysaccharides from natural sources.

Method	Mechanism	Range of Application	Target Production Properties	Advantages	Disadvantages
Fractional precipitation	The solubilities of polysaccharides are different in different solvents.	crude polysaccharides with different molecular weight distribution	obtain polysaccharides of different molecular weights	① simplicity of process ② can obtain polysaccharides with different molecular weight distributions.	① easy to produce co-precipitation ② low efficiency
Cetyltrimethylammonium ammonium bromide (CTAB) precipitation	Long chain quaternary ammonium salt can form complex with acidic polysaccharides or long chain polysaccharides and then precipitate.	most crude polysaccharides, especially the acidic polysaccharides.	obtain acidic and neutral crude polysaccharides	① low cost ② simple equipment requirements	① the great destruction of structures of polysaccharides ② low yield
Metal complexation	Polysaccharides can be complexed with specific ionic compounds to precipitate.	most crude polysaccharides	obtain free polysaccharides of different properties	① simplicity of operator ② low cost	uneasy to control the degree of reaction leading to the irreversible changes of structures of polysaccharides
Anion-exchange chromatography	It is the same as ion-exchange with reversible adsorption and bond adsorption.	acidic, neutral, and viscous polysaccharides, especially complex polysaccharides that bind to proteins	obtain homogenous polysaccharides	having a large separation capacity and satisfactory effects	① high cost ② The flow rate of eluent is easily affected by the changes of volumes, which is sensitive with the changes of eluent pH or the ion strength of solution
Gel column chromatography	molecular sieve principle, according to the size and shape of polysaccharides	most crude polysaccharides	obtain homogenous polysaccharides from different molecular weight ranges	quick, convenient, and effective separation process	① strict conditions for separation ② unsuitable for the separation of mucopolysaccharides
Affinity column chromatography	molecular affinity	polysaccharides with affinity to groups on chromatographic columns	obtain homogenous polysaccharides with different properties	① can separate polysaccharides with less content. ② one-time enrichment of polysaccharides is very high.	difficult to find suitable ligands
Cellulose column chromatography	molecular sieve principle	acidic and neutral polysaccharides	to obtain polysaccharides from different molecular weight ranges	polysaccharides with high purity	time-consuming, especially for the more viscous acid polysaccharides
Macroporous resin chromatography	molecular sieve and selective adsorption principle	most polysaccharides	to obtain polysaccharides from different molecular weight ranges	① high adsorption capacity ② good selectivity ③ reproducibility	the ability to separate polysaccharides with different properties is weak

**Table 2 molecules-24-03122-t002:** Summary of chromatography application in natural polysaccharides.

Order	Chromatography	Column Type	Elution Condition	References
Uniformity and molecular weight determination				
1	HPGPC	Shodex sugar KS-805+ guard column KS-G (300 × 7.8 mm)	unknown	[73]
2	HPGPC	TSK-gel column G4000PW_XL_	ultra-pure water containing 0.1% (*w*/*w*) NaN_3_, 0.5 mL/min	[74]
3	HPLC	TSK G4000PW_XL_ (300 × 7.8 mm)	deionized water, 0.6 mL/min	[75]
4	HPLC	TSK-gel G4000PW_XL_ (300 × 7.8 mm)	0.003 mol/L CH_3_COONa solution, 0.8 mL/min	[76]
5	HPSEC	TSK-gel G4000 PW_XL_ (300 × 7.8 mm)	distilled water, 0.6 mL/min	[77]
Monosaccharide composition analysis				
1	GC	HP-5 fused silica capillary column (30 m × 0.32 mm × 0.25 mm)	N_2_ carrier gas, 1.0 mL/min.	[78]
2	HPLC	Waters xbridge-C18 column	0.1 mol/L potassium phosphate buffer, 1.0 mL/min	[79]
3	HPAEC	CarbopacTM PA-20 column (4 mm × 250 mm)	unknown	[80]
4	HPGPC	L-aquagel-OH 40 pre-column (300 × 7.5 mm) PL-aquagel-OH 30 separation column (300 × 7.5 mm)	NaAc solution (0.003 mol/L), 1.0 mL/min	[81]
5	GC-MS	polycarborane–siloxane capillary column (25 m × 0.22 mm i.d. ×0.1 µm film thickness)	Helium carrier gas, 1 mL/min	[82]
Polysaccharide separation and purification				
1	Anion exchange column chromatography and gel column chromatography	DEAE-Cellulose A52 (2.6 × 30 cm) Sephadex G-100 gel filtration column (1.6 × 70 cm)	NaCl aqueous solution (0–1 mol/L)/Deionized water, 9 mL/h	[83]
2	Anion exchange column chromatography and gel column chromatography	anion-exchange chromatography column (2.6 × 37 cm) DEAE-Sepharose chromatography column	NaCl (0–0.6 mol/L), 4 mL/min	[84]
3	Anion exchange column chromatography and gel column chromatography	DEAE-52 cellulose gel (2.5 × 60 cm) Sephadex G-100 column (1.5 × 100 cm)	NaCl (0–0.3 mol/L)	[85]
4	Gel column chromatography	DEAE-52 cellulose chromatography column (1.6 × 60 cm)	NaCl (0–0.3 mol/L), 0.64 mL/min	[86]

Abbreviations: HPGPC, high-performance gel permeation chromatography; HPSEC, high-performance exclusion chromatography; HPAEC, high-performance anion exchange chromatography.

**Table 3 molecules-24-03122-t003:** Summary of monosaccharides composition, molecular weight, main structure, biological activity, and reference list of natural polysaccharide in recent five years.

Name	Material Source	Monosaccharide Composition and Proportion	Molecular Weight	The Main Structure	Biological Activities	References
AAPS-1	*Acanthophyllum acerosum* roots	Glc, Gal, Ara in a ratio of 1.6:5.1:1.0	23.2 kDa	→6)-α-d-Galp-(1→residues	anti-oxidation	[87]
PRG	*Phellinus ribis*	unknown	5.16 kDa	β-d-glucan containing a (1→3) linked backbone	neuroprotection	[88]
EGSP	*Gleditsia japonica var. delavayi* seeds	unknown	1913 kDa	β-1,4-d-mannopyranose	unknown	[89]
CSPS-1	*Cordyceps sinensis*	Glc, Gal, Xyl, Man, Rha in the ratio of 30.67:13.37:5.40:2.39:1.0	11,700 kDa	(1→6)-linked α-d-Glc and α-d-Gal	antitumor	[90]
GP90-1B	*Psidium guajava* fruits	Glc, Ara in a molar ratio of 9.92:84.06	unknown	(1→5)-linked-α-l-arabinose, (1→2,3,5)-linked-α-l-arabinose and (1→3)-linked-α-l-arabinose	anti-oxidation	[91]
LRLP4-A	*Lycium ruthenicum* leaves	unknown	135 kDa	1→6-linked β-galactopyranosyl residues substituted at *O*-3 by arabinosyl or galactosyl residues	immunomodulatory	[92]
GP70-2	*Psidium guajava* fruits	D-Gla, L-Ara in a molar ratio of 1:1	74 kDa	(1→3) linked α-l-Ara and (1→3,6) linked β-d-Gal	anti-oxidation	[93]
ZCMP	*Zostera caespitosa*	GalA, Api, Gal, Rha, Ara, Xyl, Man in a molar ratio of 51.4:15.5:6.0:11.8:4.2:4.4:4.2	77.2 kDa	70% AGA (α-1,4-d-galactopyranosyluronan), 30% RG-I, (→4GalAα1,2Rhaα1→, with a few α-l-arabinose)	anti-angiogenesis, immune regulation	[94]
IPSII	*Isochrysis galbana*	Glc, Gal, and Rha	15.934 kDa	a β-type heteropolysaccharide with a pyran group	anti-oxidation	[95]
CPTC-2	*Taxus chinensis* leaves	Glc, Man, Xyl, Ara, Rha, Gal in a molar ratio of 1.00:0.32:0.27:3.34:1.22:1.84	73.53 kDa	α-(1→3) Araf, α-(1→5) Araf and α-(1→4) Galp with branches composed of α-(1→3,5) Araf and β-(1→3,6) Manp	antitumor	[96]
GFP	*Grifola frondosa* fruits	Rha, Xyl, Man, Glc in a molar ratio of 1.00:1.04:1.11:6.21	155 kDa	every→3)-Glcp-(1→and one→3,4)-Glcp-(1→connected interval with a small amount of 1→, 1→4,1→6 glycosidic linkage	immunomodulatory	[97]
TC-DHPA4	tissue-cultured *Dendrobium huoshanense*	Rha, Ara, Man, Glc, Gal, GlcA in a molar ratio of 1.28:1:1.67:4.71:10.43:1.42	800 kDa	→6)-β-Galp-(1→6)-β-Galp-(1→4)-β-GlcpA-(1→6)-β-Glcp-(1→6)-β-Glcp-(→	unknown	[98]
*DQW1Pa1*	*Daedalea quercina*	D-Mannose, D-Glccose, D-Galactose, D-Xylose, L-Fucose, L-Arabinose and L-Rhamnose	16 kDa	1-3 linked linear glucose backbone	unknown	[99]
HM_41_	aerial part of *Halenia elliptica*	Rha, Ara, Xyl, Man, Gal, Glc in a molar ratio of 1.0:5.5:1.8:3.0:9.4:21	11.7 kDa	β-(1→4) Gal, β-(1→4) Glc and b-(1→6)Glc.β-(1→4)Gal	anti-oxidation	[100]
ACP1-1	*Anredera cordifolia* seeds	Man, Glc, Gal in a molar ratio of 1.08:4.65:1.75	46.78 kDa	(1→3,6)-galacturonopyranosyl residues interspersed with (1→4)-residues and (1→3)-mannopyranosyl	unknown	[101]
GalM	*Sesbania cannabina* seeds	unknown	1420 kDa	β-1,4-d-mannan	antitumor	[102]
LPR	*Lilium davidii var. unicolor* Cotton roots	Glc, Man in a molar ratio of 2.9:3.3	51.2 kDa	beta-(1→4)-linked D-glucopyranosyl and beta-(1→4)-linked D-mannopyanosyl	anti-oxidation	[103]
EPS	a newly isolated psychrophilic Antarcticfungus *Thelebolus*	unknown	unknown	(1→3)-linked β-d-Glccan backbone with (1→6)-linked branches of β-d-Glccopyranosyl units	antitumor	[104]
ASPP	purple sweet potato	Rha, Ara, Xyl, Man, Glc in the molar ratio of 2.8:1.9:1.0:7.6:53.3	18 kDa	1,4-linked Glcp with side chains attached to the O-6 position	anti-inflammatory	[105]
CP-III	*Cyclocarya paliurus* leaves	Gal, Ara, GalA, Rha, Glc, Xyl and Man in a molar ratio of 31.1:27.5:22.0:6.7:5.8:3.8:3.1	72.7 kDa	→4)GalAp(α1→ and →2)Rhap(α1→4)GalAp(α1→	pectin like polysaccharides	[106]
RCNP	*Codonopsis pilosula* roots	Ara, Gal in a molar ratio of nearly 3:1	11.4 kDa	arabinan region: (1→5)-linked Araf residues with side chains branched at the O-3 position, arabino galactan region: (1→4)-, (1→6)- or (1→3)-linked Galp along with small amounts of branches at the O-3 position of the (1→6)-linked Galp or O-6 position of the (1→3)-linked Galp residues	immunomodulatory activity	[107]
EPS-2	Saffron	unkown	40.4 kDa	(1→2)-linked -d-Manp, (1→2, 4)-linked-d-Manp, (1→4)-linked-d-Xylp, (1→2, 3, 5)-linked-d-Araf, (1→6)-linked-d-Glcp with-d-Glcp-(1→and-d-Galp-(1→as sidegroups	protection of cochlear hair cells from ototoxicity exposure	[108]

Abbreviations: Ara, arabinose; Xyl, xylose; Man, mannose; Glc, glucose; Gal, galactose; Fuc, fucose; Fru, fructose; Rha, rhamnose; Sor, sorbose; Tal, talose; GlcA, glucuronic acid; GalA, galacturonic acid; Galp, galactopyranose; Glcp, glucopyranose.

**Table 4 molecules-24-03122-t004:** Raman spectrum of polysaccharide functional groups.

Raman Spectrum/cm^−1^	Group/Atom
350–600	Pyranose ring
600–950	Heterocarbon model
950–1200	Glycosidic bond type
1200–1500	CH_2_ and C-OH deformation

**Table 5 molecules-24-03122-t005:** Common methods for determination of polysaccharide primary structure.

Number	Polysaccharide Structure Analysis and Determination Project	Common Method
1	Overall structural analysis Homogeneity and molecular weight determination	High performance gel permeation chromatography (HPGPC), osmotic pressure, light scattering, viscosity, polypropylene gel electrophoresis, etc.
2	Overall structural analysis Monosaccharide composition and proportion	Complete acid hydrolysis, HPLC, GC, GC-MS, high performance ion chromatography, etc.
3	Glycoside ring form	Raman spectroscopy such as infrared spectroscopy.
4	Glycosidic linkage sequence	Selective acid hydrolysis, sequential hydrolysis of glycosidase, nuclear magnetic resonance, etc.
5	Hydroxyl substitution	Methylation, periodate oxidation, Smith degradation, GC-MS, nuclear magnetic resonance, etc.
6	Polysaccharides chain-peptide bond linkage	Dilute alkali hydrolysis, hydrazine reaction, amino acid composition reaction, etc.
7	Amorphic form substituted by glycosides	Glycosidase hydrolysis, nuclear magnetic resonance, infrared chromatography, laser, etc.
8	Monosaccharide residue type and glycosidic linkage site	Methylation analysis, GC-MS, etc.
9	Oligosaccharide determination	Partial acid hydrolysis, GC-MS, MS, etc.

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
