# Peer review of "The Preparation and Structure Analysis Methods of Natural Polysaccharides of Plants and Fungi: A Review of Recent Development"

_molecules, 2019, doi:10.3390/molecules24173122_

Round 1

Reviewer 1 Report

This is nice educational review of the modern methods used for isolation and characterization of plant and fungal polysaccharides. Review does not deal with bacterial and animal polysaccharides. Authors need to explicitly point to the area of review, change title and mention it in the text, because now there is no indication that it deals only with plants and fungi and thus can be misleading.

Please cite more recent works of Ciucanu on methylation analysis. Now there is a reference to his first work, but he made many improvements later which nobody cite. There is strange statement that now Needs methods is most common for methylation, although Needs did not change Ciucanu (written Ciucan on line 346) procedure and there is practically no citations of his work.

In NMR section chemical shifts are given without units, please add "ppm" after numbers. Looks like authors have no experience with NMR and this section looks quite uninformative.

Author Response

Dear reviewer,

Thanks for your helpful comment on this paper and we wrote a response Letter in the attachment.

Kind regards,

Yan Ren

Aug.25,2019

Reviewer 2 Report

Dear editor,                                                                                        

The review-manuscript submitted by Yan Ren et al.  entitled: “The preparation and structure analysis methods of natural polysaccharides: a review of recent development” aims to investigate the main strategies to extract and analyse natural polysaccharides. All things considered it is an interesting study, which can be accepted for publication after major revision. Please to see my comments below to improve the revised manuscript.

Comments:

- Author should reformulate the title of this publication in the revised manuscript. This title is too confused for reader.

- Authors must modify the list of key words.

- In introduction part, authors have made statements on polysaccharides and not backed it up with last good references concerning the production and the applications of polysaccharides. What do you mean by synthesis of polysaccharides? Authors must give more information about “synthesis processes”. More, authors must give more explanations regarding Figure 2 (sources, etc.?).

General comment:

In the revised manuscript, the authors need to pay more attention to grammatical construction of sentences and spelling of sentences ! Authors must improve the quality of figures in the revised manuscript.

Author Response

(The authors gave the same response as above.)

Reviewer 3 Report

The title of the present review is very comprehensive, encompassing “preparation and structure analysis methods of natural polysaccharides”. Nevertheless, not all polysaccharides were included. For instance, the glycosaminoglycans, which are a very important class of natural polysaccharides, were not comprised. Accordingly, the authors should consider a less comprehensive title, indicating the types of polysaccharides that will be approached (Intracellular? Vegetal? Non sulfated).

Concerning the “Methods of Analysis” – important methods, commonly used for the analysis of oligo and monosaccharides were omitted, such as “fluorescence assisted carbohydrate electrophoresis – FACE”, for instance. Again, the authors should consider a better definition of their scope.

Author Response

(The authors gave the same response as above.)

Round 2

Reviewer 2 Report

Dear Editor,

Manuscript was improved in the revised form. Consequently, it could be accepted for publication in Molecules journal.

Best regards,